# Coping and support-seeking in out-of-home care: a qualitative study of the views of young people in care in England

Rachel M Hiller [1], Sarah L Halligan,[1,2] Richard Meiser-Stedman [3], Elizabeth Elliott,[1,4] Emily Rutter-Eley,[4] Tilly Hutt[1]

► Prepublication history and additional material are published online only. To view please visit the journal online (http://dx.doi.org/10.1136/bmjopen-2020-038461).

¹Department of Psychology, University of Bath, Bath, UK
²Department of Psychiatry and Mental Health, University of Cape Town, Cape Town, South Africa
³Department of Clinical Psychology, Norwich Medical School, University of East Anglia, Norwich, UK
⁴School of Psychology and Clinical Language Science, University of Reading, Reading, UK

**Correspondence to**
Dr Rachel M Hiller;
R.Hiller@bath.ac.uk

## ABSTRACT

**Objectives** Young people who have been removed from their family home and placed in out-of-home care have commonly experienced abuse, neglect and/or other forms of early adversity. High rates of mental health difficulties have been well documented in this group. The aim of this research was to explore the experiences of these young people within the care system, particularly in relation to support-seeking and coping with emotional needs, to better understand feasible and acceptable ways to improve outcomes for these young people.

**Design and study setting** This study used 1:1 semistructured qualitative interviews with young people in out-of-home care in England, to provide an in-depth understanding of their views of coping and support for their emotional needs, both in terms of support networks and experiences with mental health services. Participants were 25 young people aged 10–16 years old (56% female), and included young people living with non-biological foster carers, kinship carers and in residential group homes.

**Results** Participants described positive (eg, feeling safe) and negative (eg, feeling judged) aspects to being in care. Carers were identified as the primary source of support, with a supportive adult central to coping. Views on support and coping differed for young people who were experiencing more significant mental health difficulties, with this group largely reporting feeling unsupported and many engaging in self-harm. The minority of participants had accessed formal mental health support, and opinions on usefulness were mixed.

**Conclusions** Results provide insight, from the perspective of care-experienced young people, about both barriers and facilitators to help-seeking, as well as avenues for improving support.

## INTRODUCTION

The most common reason a child would be removed from their home and placed in out-of-home care is due to the experience of abuse and/or neglect.[1 2] In the UK, it is also the case that most young people who enter care are removed from their home when school-aged or older, meaning early experiences

## Strengths and limitations of this study

► This study is focused on young people living in out-of-home care, representing an under-researched group where there remains significant need.
► Using 1:1 qualitative interview methods, we gained an in-depth understanding of the views and experiences of these young people, in terms of coping and help-seeking around mental health.
► We used opportunity sampling methods, which may introduce sampling bias.
► As this study uses qualitative methods, we cannot conclude whether different types of reported coping might be associated with better or worse outcomes.

of maltreatment have often been ongoing for a number of years.[1] Once in care many continue to face ongoing instability. Separation from siblings is common, as are moves between different placement providers, resulting in the potential for inconsistent support.[3 4] Given these accumulative experiences, it is perhaps unsurprising that there is well-established evidence that young people in out-of-home care experience high rates of psychological difficulties compared with their peers.[5–7]

The failure to adequately address the psychological or emotional needs of young people in care has been identified as a key driver of many of the well-documented poor outcomes that can categorise this group, including school attainment and later rates of homelessness and unemployment.[8] Yet, despite the long-documented poor outcomes, there has been limited evidence of improvements in the overall picture of mental health for children in care in the UK over the past decade.[1] One concern is that these young people can fall between the gaps of children's social care and national health service provisions, which can result in poor access to

mental health services.[9 10] With resource issues affecting child and adolescent mental health services (CAMHS) in many countries, it is also largely impossible for these services to address the large number of young people in care who may benefit from professional mental health support.

In the face of potentially limited professional support services, young people will inevitably develop day-to-day coping strategies and support networks—whether perceived to be adaptive or maladaptive. Some qualitative work has investigated the views of young people in care in relation to support networks, and specifically their relationships with carers, where it has been found that developing trust, consistent boundaries, and empathy are considered important.[11 12] However, there is less evidence for how they may or may not engage with other support networks, including peers, school and social workers. Such evidence is potentially particularly important for this group, given one-third of young people in care in the UK will have more than one carer in any given year,[1] and given the well-documented cyclical association between placement breakdowns and emotional difficulties.[4 13] Beyond the role of external support networks, we know little about what strategies young people in care may be using themselves to manage their emotional well-being. While many young people in care struggle with emotional difficulties, it is important to note that not all do. Evidence from the wider child mental health field shows that engaging in certain types of coping strategies can be associated with better or worse mental health difficulties,[14 15] including in care-experienced young people.[16] Understanding how young people in care cope with emotional difficulties, particularly in the face of potentially disrupted support networks, is crucial to understanding both the experiences of this group, and the avenues for improving formal and informal support.

Using a semistructured qualitative approach, we sought to explore the following questions: (1) where do young people in care seek support for emotional difficulties, both in terms of social support and professional services?; (2) what do they view as barriers to seeking help? and (3) what coping strategies do they use when experiencing emotional difficulties? This study included a focus on more day-to-day coping and help-seeking (eg, if they felt stressed) and coping and help-seeking around more significant mental health difficulties.

## METHOD
### Recruitment and sample
We recruited 25 young people aged 10–16 years old. Participants were all part of a larger quantitative study on the mental health and well-being of young people in care.[16] We used opportunity sampling, with consecutive young people invited to participate in an optional qualitative interview alongside their participation in the larger study. For the full procedures for the larger study see Hiller et al.[16] The sample size was based on general guidance for qualitative methodology.[17 18] While clear guidance for sample sizes in qualitative research remains limited, our sample size was chosen to be in line with, or larger than, similar studies in the field.[19 20] It was also based on our previous qualitative work with young people in terms of the sample required to reach saturation, with capacity to continue recruitment if it was felt saturation was not reached or the data were inadequate (which was ultimately not the case). The sample size also allowed us to explore views of older and younger children within our age range, and include young people with a range of mental health needs, from those who were managing well to those struggling with chronic mental health difficulties.

Participants came from two local authorities in England. Based on national data, one was a smaller-sized local authority and one was medium-sized, with both in largely urban areas. Informed consent for the young person's participation was provided by their social worker, while the young person provided informed assent (or consent if 16 years old). Eligibility for participation in the quantitative study were: aged 10–17 years old, on full care order if under 16 years old (where the local authority has acquired parental responsibility for the young person), absence of developmental disability that would preclude the child from mainstream schooling and absence of significant current risk (eg, current severe suicidality). Young people could be in any style of placement, including foster placement with a non-biological carer, kinship placement or residential care home. Descriptive information on the sample is presented in table 1.

### Procedure and measures
In-person 1:1 interviews were conducted by two researchers. The lead researcher (RH) had prior training and experience in qualitative research, while the other (EE) was experienced researching with vulnerable young people and adults and received further training specifically on qualitative procedures. We used a 1:1 semistructured interview format, rather than focus group, due to the potential for sensitive information being discussed that young people might not have wished to disclose in a

**Table 1** Sample demographics

| Young person descriptives (N=25) | |
| --- | --- |
| Age, M (SD) | 13.0 (2.24) |
| Sex (% female) | 56% |
| Ethnicity (% Caucasian) | 80% |
| Age entered care, M (SD) | 8.7 years (4.2) |
| Time in care, M (SD) | 4.2 years (3.6) |
| % who had multiple placements | 80% |
| Type of placement | |
| Non-biological foster carer | 76% |
| Biological foster carer (eg, grandmother) | 16% |
| Residential care home | 8% |

group. For one of these interviews two siblings requested to complete the interview together, while for two interviews the carer remained present at the request of the young people. All others (n=22) had only the young person and interviewer present. RH and EE met regularly throughout the study to monitor the quality of the interviews and reflect on the content. All interviews were audio recorded and later transcribed verbatim. As a thank you for volunteering their time, participants each received a £10 voucher. As per standard risk management and safeguarding procedures, where appropriate, summary letters were provided to social workers to support potential referrals to mental health services.

### Descriptives and semistructured interview

Basic descriptive information was collected from the young person's social worker, as part of the larger quantitative study. This included the age and reason that the young people had entered care. The semistructured interview was developed based on the literature on qualitative interviewing techniques and qualitative and quantitative work in the child mental health field.[9 21 22] Interviews broadly covered four core areas: (1) views on what is broadly helpful and unhelpful within the care system, (2) coping and support for children's mental health and well-being (including at home, within social care and experiences around mental health services), (3) coping and support for trauma-related mental health more specifically (eg, how young people cope with memories of their maltreatment), and (4) views on ideal support. Questions were primarily focused on the young person's subjective experiences (see online supplemental appendix 1 for interview schedule). Additionally, participants were also asked what advice they might give a friend in a similar situation, to elicit broader information from young people who might have struggled to answer these questions about themselves. Questions were primarily open-ended (eg, 'If you're having a hard time what do you do to make yourself feel better or different?'). Some direct prompting and questioning were used to clarify comments and gain further insight (eg, 'Was that helpful?'). Interviews lasted from 6 to 32 min (M=15.36 min).

### Data analyses

Transcripts were analysed using a reflexive thematic analysis approach[23 24] in NVivo V.12. Given the exploratory nature of the study, thematic analysis was chosen as the data analytical approach as it allows for a detailed exploration of the data and does not rely on a specific theoretical framework. Coauthor (ER-E), who was not involved in any of the interviews or the broader quantitative study, coded all transcripts. While the purpose of qualitative research is not to completely remove or minimise the influence of researcher subjectivity, a coder who had not been involved in study design or data collection was used, to reduce potential biases. She first read all transcripts to gain an understanding of the overall data (data immersion) and then systematically coded each transcript using

an iterative process.[24] Following the systematic coding of all transcripts, codes were grouped to form overarching themes. The broader themes were also separated into subthemes, to accurately capture more fine grain information provided by participants and ensure the themes appropriately reflected different viewpoints. Coauthor (TH) independently explored all (uncoded) transcripts and generated their own key themes. A consensus meeting showed strong agreement between the primary themes. There was not capacity within the study to seek further input from participants at this point, however reflective practices were used throughout all interviews to ensure clear understanding of the discussion and to minimise any chance of misinterpreting what was being said (eg, reflecting, summarising back to participants and seeking further clarification where needed).

### Patient and public involvement

This study occurred as part of a series of projects, which involves extensive consultation with services, as well as feedback from care-experienced young people and foster carers. This feedback particularly shaped the research questions and use of qualitative methods. Young people and carers were not involved in recruitment for the project. There has been wide written and verbal feedback to stakeholders as part of the larger project.

## RESULTS

### Descriptive information

The sample comprised of 14 females and 11 males, aged 10–16 years old. They had entered care between infancy and 16 years old, and had been in care from 6 months to almost 14 years, while 80% of the sample had had more than one placement since entering care. Most (76%) were living in a foster placement with a non-biological carer, while four were in a kinship placement (eg, with a grandmother) and two lived in residential care homes. All were in care due to the experience of abuse and/or neglect.

### Interview themes

Thematic analysis identified three primary themes covering the broader experience of being in care (theme 1); the centrality of social support to well-being and mixed views on professional help (theme 2); and the use of both adaptive and maladaptive day-to-day coping strategies (theme 3). A list of themes and subthemes can be found in table 2.

### Theme 1: being in care could create a sense of safety but some young people struggled with a lack of control and agency

When asked what it was like to be in care, most, but not all participants said that being in care was a better or safer alternative to living with their biological family and that they felt safer in care. However, almost all discussed

**Table 2** Themes and subthemes from thematic analysis

| Themes | Subthemes |
|---|---|
| Being in care could create a sense of safety but some young people struggled with a lack of control and agency | Positive aspects of being in care were the sense of safety and being part of a family. |
| | Negative aspects of being in care included feeling different and not being listened to or respected. |
| Social support was considered key to coping but not always available, while access to professional support was inconsistent. | Day-to-day support from a trusted adult was identified as very important for well-being. |
| | Formal mental health support was often unavailable and opinions on usefulness were mixed. |
| Talking about problems was seen as important, but maladaptive coping strategies were also reported | Talking about worries or challenges was identified as a helpful strategy for coping with negative emotions. |
| | A majority avoided talking about their pre-care experiences and in the context of mental health difficulties maladaptive coping was common. |

occasions of feeling stigmatised, as well as experiences of being poorly supported by services.

### Subtheme 1: positive aspects of being in care included a sense of safety and being part of a family

The majority of participants reported that feeling safe and secure was a particularly positive aspect of being in care. Some participants also described experiencing a sense of relief when first coming into care, recognising that it was not appropriate for them to remain in an unsafe environment.

> Relieved actually from going to a bad environment into somewhere I thought could be quite safe. (16 years old)

> To know at the end of the day you can go home, instead of thinking 'OK, I'm going to go home tonight, am I going to find my mum crying on the floor again drunk?' (12 years old)

The majority of young people discussed how they have felt 'normal' and like part of a family since coming into care. This was expressed regardless of participants' current relationships with biological family, with many describing strong and important bonds with both their biological and foster family. Feeling like part of a family was especially valued and some expressed their appreciation for the support and sense of belonging provided by their foster carers.

> Just feel like an ordinary family really, like an ordinary person. Sometimes I don't even realise I'm in care. (11 years old)

> I'm lucky I ended up here. I'm so goddamn lucky I ended up here, coz this is my home now, it's my family and I love 'em, I love 'em, I actually do, coz they're my family. (12 years old)

In some cases, these positive aspects of their placement were reported to facilitate improvements within different areas of participants' lives, such as their school, and their ability to cope with difficult situations and emotions.

> They [carers] like understand what I'm saying and like if I'm annoyed they help me to know how to calm down. (10 years old)

> I started knuckling down at school, because I felt safe. (12 years old)

### Subtheme 2: negative aspects of being in care included feeling different and not being listened to or respected

Most participants discussed feeling 'different' because they were a young person in care, and were concerned about others' judgement. Mandatory forms and compulsory meetings with social workers could create difficult situations for participants, or 'annoying' disruptions, which served as regular reminders that they were different from other children, who were not asked to fill in forms and (for example) leave class to meet with a social worker. Alongside this, concerns about others' (eg, teachers, peers) lack of understanding meant that many participants were very conscious of being judged or stigmatised, particularly within the school context.

> Most people think you are naughty 'cause you're in care. (10 years old)

> Like the social workers coming in here [school] or whatever... it's awkward 'cause like a normal person wouldn't really have that. (10 years old)

Among adolescents and particularly those struggling with mental health difficulties, feeling different from their peers was often accompanied by complaints of lacking freedom and control, generating a feeling of restricted independence. This led some older participants to express their desire to leave the care system and move into independent living, in order to take back control over their lives.

> I have no control over my own life. (16 years old)

I want to go into independent living, because I've had enough of living with, like, parent figures and I just want to be on my own and be independent and prove that I can do it. (16 years old)

### Theme 2: social support was considered key to coping but not always available, while access to professional support was inconsistent

The majority of participants most valued informal day-to-day support from a trusted adult, while describing formal/professional support to be less available and less helpful. Many participants expressed the need for professional support to be improved and made more readily available.

### Subtheme 1: day-to-day support from a trusted adult was important for well-being

Having a trusted adult to talk to about day-to-day problems or concerns, whenever they felt they needed to, was identified as the most valuable form of support for participants. Particular aspects valued in a trusted adult were their ability to listen, to 'keep secrets' and to 'give good advice'. The importance of being able to call on this support as and when it was needed was often emphasised. For example, participants described the benefits of being able to talk to carers after a bad day or to speak with key staff at school if there was an argument in a lesson. Knowing there was always a trusted adult available to speak to was central to maintaining well-being for many participants.

Telling [your worries/concerns to] someone you trust makes you a lot calmer. (16 years old)

Overall, carers were by far reported as the primary source of support from the perspective of most of the participants. Some participants also talked about their social worker or a school staff member being key sources of support. For older teens, some also mentioned close friends as their key source of support, particularly if they viewed friends as more trustworthy than adults.

…most foster carers are very nice and they really help you with your life… they look after and like you. (10 years old)

Whenever something [is] tough, I'll tell her [social worker] and she'd always take me out and then we'd always chat about something different and then she'd take my mind off it. (16 years old)

My school nurse helped me a lot because whenever I had a problem I'd go to her and she'd try to sort it out as quick as possible. (16 years old)

Really close friends, I can trust them with my life. (15 years old)

While there were some cases of very positive relationships between a young person and their social worker, perceived ineffective communication between social workers and young people was identified as one of their central problems in terms of feeling supported. A focus of many interviews was how transient their social worker was, with almost all having multiple different social workers over their time in care, or even over the past year. This transience made it particularly difficult for young people to build trust, which could also influence their willingness to disclose information about their early experiences and their general trust of professionals. Some also described situations where they believed that their confidentiality had been breached by adults in their support network, further resulting in them feeling let down and their trust being eroded.

I just want one […] one person [Social Worker] who is going to be there for quite a long time and not going to leave all of a sudden and I can build up a bond. (15 years old)

You tell them [about your experiences] and then they leave and you can't tell them anymore. So you're like oh my God who can I tell? Who can I trust? It's hard. (16 years old)

A minority of participants perceived that there was no one they could talk to. These participants were all adolescents and all reported struggling with significant mental health difficulties, including engaging in self-harm and/or drug or alcohol use. For these young people, the lack of a trusted adult, which largely included an unstable placement, was particularly problematic. Perceived unhelpful foster placements were discussed in terms of young people perceiving carers fostering 'for the money', blaming or not trusting the young person, not putting effort in to spending time with them, or the young person perceiving that the carer had not been appropriately trained.

Since I was seven all I've ever done is either sat on the sofa and watched tv on my own coz no one talks to me or I've sat in my room and watched tv on my own because nobody ever talks to me. (16 years old)

### Subtheme 2: formal mental health support was often unavailable and opinions on usefulness were mixed

Formal support from mental health professionals was identified as being often unavailable and unhelpful. For those who had seen a mental health professional, this was often the school counsellor, with the minority accessing CAMHS. Among those who had received support, either in the form of a school counsellor or CAMHS, opinions were mixed, with a slight majority describing this support as unhelpful. Young people had difficulty specifically identifying what they found unhelpful about this support, although some did describe feeling restricted to a scheduled appointment and having to talk about their past despite not wanting to do so.

It made it worse like, having a scheduled time to go and talk about things when I don't really want to talk about them, like I'd rather just say it when I felt like

saying it. It just made me think about it more which made it worse. (14 years old)

Similarly, those who had positive experiences with mental health professionals struggled to identify what it was that they had done that had been helpful, although some of them did highlight that practitioners had provided them with a safe space to talk and help them cope.

It made me talk to people and like actually not just keep everything to yourself. (15 years old)

I got my feelings clear to everyone. (11 years old)

### Theme 3: talking about problems was seen as important, but maladaptive coping strategies were also reported

When participants were asked what they did when experiencing negative emotions, a range of coping strategies were identified. Some of these strategies were adaptive, such as talking to somebody they trusted, and some were maladaptive, such as self-harm.

### Subtheme 1: talking about worries or challenges was identified as a helpful strategy for coping with negative emotions

Talking about day-to-day worries or problems with a trusted person was identified as a common strategy used in response to negative emotions, such as sadness, anger or anxiety. As previously highlighted, carers, teachers and social workers were among those who participants felt comfortable speaking to. This was discussed as a primarily helpful strategy, often resulting in feeling better afterwards or enabling problems to be resolved.

If you're feeling down, always talk to someone that you're close to about it, never hold it in. (16 years old)

### Subtheme 2: a majority avoided talking about their pre-care experiences and in the context of mental health difficulties maladaptive coping was common

While talking to someone about current worries was seen as important for coping, most participants thought it was best to avoid talking about their pre-care experiences. Indeed, when asked what advice they might give a friend who was struggling, many endorsed they would encourage them to talk to someone, but then also endorsed that this was not something they themselves did.

Speak up and tell people how you are feeling and what you've been through, and that you're struggling to cope. Because if people don't know people can't help.

Interviewer: Did you do that?

No, because I didn't know them and I don't trust people I don't know. (16 years old)

Reasons for avoiding discussing their maltreatment varied but the most common justifications were either because talking about pre-care experiences made them feel uncomfortable or because they did not think it would

be helpful. This sense of discomfort when talking about previous experiences often appeared to be associated with a lack of trust in adults and services. Participants also discussed feeling embarrassed, uncomfortable with the amount of attention placed on them and unsure of how to act when describing events.

I don't really like talking about my past too much, it's, it's like all attentions on me and it's like awkward. (14 years old)

It don't change nothing, it don't mean it go back. It still happened you can't make it [go away]. (10 years old)

From those who identified talking as an unhelpful strategy, many held the attitude that, as events were in the past, talking about them would not change anything. Some also felt that such discussion acted as a barrier to moving forward. This was particularly reported by young people who were experiencing serious mental health difficulties, including current distressing intrusive memories of their maltreatment.

I'm not saying 'suck it up and be a man' but just try and just keep your mind off it, keep doing what you're doing cos you don't really want to let the past like take over you. Because the past is in the past and I really don't want to bring my past up. I'm trying to move forward for the minute but obviously if people are bringing it up, I can't. (15 years old)

In general, many participants endorsed trying not to think about their negative (eg, maltreatment) memories and avoidance of talking about their past as a primary way to cope. In most cases this involved distracting themselves with activities, such as watching TV and spending time with friends. Indeed in many cases when talking about their support systems, young people endorsed that adults helping to distract them was helpful (eg, changing the topic; social workers taking them to lunch; see theme 2 quotes). These distractions shifted the focus of their attention onto something else, allowing them to avoid thinking about their early experiences.

[in response to question about coping] just watch tv or something. (10 years old)

I would go on the PlayStation and just kill people. (15 years old)

In the case of adolescents who were struggling with mental health difficulties, there were more frequent examples of destructive coping strategies, particularly self-harm and drug and alcohol use. Many of these young people reported that they did not have any other way of coping with their emotional difficulties.

I pretend to be someone I'm not. I pretend to be okay when I'm not and let emotions build up until eventually I just break. (12 years old)

I cry and I hurt myself that's literally the only way I know how to deal with things anymore. (16 years old)

## DISCUSSION

This study aimed to learn more about the views of young people in out-of-home care, in relation to coping, support and help-seeking with emotional difficulties. Across the interviews there were core common themes, including the importance of having a consistent and trusted adult with whom they could share their worries. Carers were identified as the primary source of emotional support by most young people. While many participants talked about the importance of talking to your support system as a means of coping, most also thought that it was not helpful to talk about their pre-care experiences. Adolescents experiencing mental health difficulties also commonly struggled to identify adaptive coping strategies or support systems.

Similar to other qualitative work with young people in care,[9 12] participants viewed the support of a trusted adult as key to coping with daily stress and interpersonal difficulties, and with more pervasive mental health difficulties. In the vast majority of cases, the carer (or a key worker if in residential care) was viewed as the primary source of this support, although some young people talked about this support also coming from their social worker, a trusted staff member at school, or a friend. Trustworthiness was discussed as particularly central to the young person feeling supported, including the perception that the person would be able to keep information confidential. A lack of trust was a key concern and potential barrier to seeking help. While some young people had positive relationships with their social workers, many participants discussed changes in social workers as another key barrier to building a trusted relationship and seeking support for emotional difficulties. This reflects similar findings by Wood and Selwyn in their qualitative work, where young people in care also discussed the challenges of navigating regular changes in social workers and the ultimate perceived negative impact on their subjective well-being.[25] Findings highlight the need for services to explore policies around mitigating some of the consequences of social worker changes on the young person's well-being. Social work has some of the highest burn-out and turn-over rates of any profession. Providing more robust training and support to social workers is considered key to reducing social work changes.[26] However, given some change is inevitable, policies for how these changes happen are crucial, to minimise the potential eroding of trust. Given the age range included adolescents, it was interesting that peers were less commonly identified as a primary source of support. In adolescence, research has shown that friendships can be central to resilient functioning.[27] The lack of focus on peers may be due to the concern young people in care expressed about being stigmatised by peers for being in care, meaning they are less likely to turn to friends. It may also be that some, especially teens

experiencing mental health difficulties, struggled in school in general and had changed schools during their time in care or were no longer in school, potentially influencing the opportunity to form trusting friendships. The role of friendship in promoting resilience in this group of young people is worthy of further exploration.

While many participants talked about going to their carer for support if they were feeling worried or upset (eg, if they were upset about something at school), many at the same time emphatically stated that they did not think it was useful to talk about their maltreatment experiences, with either their carer or mental health professionals. While this was the case across much of the sample, it is perhaps most challenging for those young people who discussed experiencing significant mental health difficulties. This subset of the sample most commonly reported feeling that they had no person they could turn to for support, that they did not want to talk to a professional about their maltreatment or mental health, and had no strategies for coping with their mental health beyond self-harm and drug and alcohol abuse. Beliefs around not talking about maltreatment or trauma present a particular challenge to mental health and social care professionals. Of course, young people could and should not be forced to talk about traumatic experiences. Some may also be coping fine and not need to talk through their experiences. It is also completely understandable that young people may not want to talk about distressing early experiences. Yet, avoiding talking or thinking about traumatic experiences (ie, thought suppression) is a key maintainer of poor mental health outcomes,[15] including for young people in care.[16] Further, a key component of evidenced-based trauma-focused psychological support involves processing the traumatic experiences, commonly via a trauma narrative.[28] There remains an urgent need to understand how services can best engage young people in care who may benefit from professional support in treatments that might involve talking about their experiences, particularly for a group where developing trust and a strong therapeutic relationship is essential.[10] This may be partly achieved by having systems, including at a social care and education level, where staff are appropriately trained and supported in engaging with these challenging conversations with young people from earlier on in their time in care, so young people know that it is safe to talk about these experiences if they want to. Many services in the UK would report using 'trauma-informed' approaches to their practice,[29] which would include creating safe spaces for young people to talk about their experiences. However, the evidence base for how these approaches are genuinely implemented and their effectiveness, remains scarce. Relatedly, many young people reported feeling stigmatised by being in care, including by professionals (eg, teachers) and peers—a finding that is also in line with other qualitative work with this group.[26] It will therefore also be important for services to consider how to address issues of stigma, which can be a barrier to help-seeking. Strength and relationship-based

approaches may be useful here,[30] but remain poorly researched in terms of how they are used in practice or their general effectiveness. These remain important areas for future research, to focus not just on targeting psychopathology but more broadly improving the emotional well-being of these young people.

For the minority of participants in this study who had received professional mental health support, including via CAMHS and school counselling, views on this support were mixed. In their 2008 review of qualitative literature on children in care and mental health services, Davies and Wright highlighted that many young people felt wary and unsure of professional mental health support, and they highlighted the importance of developing trust.[9] Ten years later, the current study found that young people in care are still experiencing these same concerns. There remains an urgent need to improve understanding of the service-level barriers to accessing professional support, including access to evidence-based psychological interventions (eg, see review [31]). Within the current UK CAMHS system, thresholds for professional mental health support often mean young people are in crisis, with complex mental health needs, by the time they can access support (eg, see [32]). Exploring feasible but appropriate avenues for providing effective evidence-based support, before reaching crisis, is clearly an important avenue for addressing the mental health needs of these young people.

## Limitations

While this study has key strengths, including a focus on an under-researched group, and the use of qualitative interviews to provide an in-depth exploration of their views on coping and well-being, findings should also be considered in light of key limitations. First, we used opportunity sampling methods, which potentially creates sampling bias. All young people were participants in a larger quantitative study of mental health and well-being.[16] Young people who did not want to talk about their experiences and mental health would likely have declined participating in this study. Participants also all came from two local authorities in England. While findings cannot necessarily be generalised to other young people in out-of-home care, it is worth noting the significant overlap between these findings and qualitative work with young people in care and foster carers in the UK, USA and Australia.[9 19] Finally, quantitative research is needed to provide more robust evidence on the consequences of different support systems and coping strategies in relation to the mental health of these young people.

## CONCLUSIONS

In sum, while there were positive aspects of being in care and many young people felt well supported by their carer, results of this qualitative study identified key areas to be addressed to improve support for the emotional well-being of this group. This included improving the continuity of social workers and carers to allow young people to develop trusted relationships with key adults, targeting broader concerns around stigma and trust (including in the school system) that may act as a barrier to help-seeking, and ensuring young people are receiving appropriate evidence-based psychological support in a more timely manner, before reaching crisis point. Addressing these issues remains a significant challenge for services, particularly alongside long-term underfunding of social care and mental health services. Yet, failing to provide adequate support, both in terms of informal day-to-day support and professional services, can have a substantial impact on the trajectory of these young people.

**Acknowledgements** The authors wish the thank the services and carers who were involved in this project and have continued to be so willing to explore practice and ways to improve support for young people in their care. We also thank the young people who participated in this study and those who have helped to shape our research. We are so grateful to them all, for their willingness to allow us to learn from their (often very difficult) experiences and understand how we can and must do better.

**Contributors** RH is the lead researcher on the project and principal investigator on the ESRC grant. She led the conceptualisation of the project, conducted a proportion of the interviews and was responsible for overall project management. SLH was involved in the conceptualisation of the project, including the design of the interview schedule, and also contributed to discussion on the themes and feedback on the manuscript. RM-S was involved in the conceptualisation of the project and contributed to the discussion about the themes and feedback on the manuscript. EE was the research assistant on the project and conducted a proportion of the interviews, and contributed to transcribing, data analysis and feedback on the manuscript. ER-E was involved in the transcribing, contributed to data analysis and provided feedback on the manuscript. TH contributed to transcribing and feedback on the codes and themes.

**Funding** This research was funded by an ESRC Future Leader Grant (ES/N01782X/1) and University of Bath Prize Fellowship awarded to RH.

**Competing interests** None declared.

**Patient consent for publication** Participants provided informed consent/assent for their participation in the study, including the use of anonymised quotes in publications.

**Ethics approval** Ethical approval was granted by the University of Bath Psychology and Social Care research ethics committees, along with permission from the participating local authorities.

**Provenance and peer review** Not commissioned; externally peer reviewed.

**Data availability statement** Data are available upon reasonable request. Anonymised transcripts will be available on request on a case-by-case basis.

**ORCID iDs**
Rachel M Hiller http://orcid.org/0000-0002-4180-8941

Richard Meiser-Stedman http://orcid.org/0000-0002-0262-623X

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
