## [Reviewer comments · BMJ Open]

ARTICLE DETAILS

TITLE (PROVISIONAL)	Coping and support-seeking in out-of-home care: A qualitative study of the views of young people in care in England
AUTHORS	Hiller, Rachel; Halligan, Sarah; Meiser-Stedman, Richard; Elliott, Elizabeth; Rutter-Eley, Emily; Hutt, Tilly

VERSION 1 – REVIEW

REVIEWER	Fiona Robards The University of Sydney
REVIEW RETURNED	04-Apr-2020

GENERAL COMMENTS	Thank you for the opportunity to review the manuscript Coping and support-seeking in out-of-home care: A qualitative study of the views of young people in care in England. The study aims to explore the experiences of young people within the care system, particularly in relation to support-seeking and coping with emotional needs, to better understand feasible and acceptable ways to improve outcomes for this group. Abstract The first line, 'Young people who have been removed from their family home and placed in out-of-home care have often experienced maltreatment', could have more impact. The first line of the article is much better: 'The most common reason a child would be removed from their home and placed in out-of-home care is due to the experience of abuse and/or neglect' Key words Suggest revision to include youth or adolescent, help-seeking. Strengths and Limitations of the Study Could be strengthened. What does this study add? Introduction The final paragraph should be moved to methods. Also, can you please clarify if the participants were asked about their own experience or their views about young people generally. Replace the final paragraph with the research question. Methods Can you be more specific about the decisions guiding sample size? Page 6 line 29: 'urban' does not need to be capitalised. 'Researcher characteristics and reflexivity' is listed as being included on page 7. This content could be much clearer. Results The results seem to be very surface level (lacking depth) – relating to examples of positives and negative. Discussion It would be useful to consider strength-based and trauma-informed approaches to working with young people in care. The
--

	recommendations put forward by the authors seem at odds with what was recommended by participants. Page 20, line 20. 'Addressing these issues remains a significant challenge, but the current work found failing to provide adequate support, especially for those with the most significant mental health difficulties, had major consequences for the young person's wellbeing'. I'm not sure that the study supports this closing statement. Overall comments While this group is under-researched, this study lacks depth in the results and entertains a particular lens of interpretation which was contrary to the views of participants.
--	---

REVIEWER	Bernard Gallagher University of Huddersfield, England
REVIEW RETURNED	13-Apr-2020

GENERAL COMMENTS	The subject matter of this manuscript (MS) is important and interesting. Some of the findings are familiar, having been covered in previous literature, but they are no less important for this. This MS is focused upon a highly vulnerable group and it is vital to have current data on the welfare of these young people and the services they receive. I think the MS is well written, being clear and concise, and it is well structured. I had only one substantial criticism and that concerns the Discussion. I thought there was a bit too much of the findings in this section. There may be a bit of repetition from the Findings section or indeed new material that should have been in the Findings section. At the same time, I thought the Discussion section could have analysed more of the wider literature. I had a number of smaller points:  1. 'Local authority/ies' and 'urban' are not proper names and should not be written with capitals. 2. The term "full care order" should be briefly explained for international readers. 3. I think "stake-holder" should not be hyphenated but should be a single word. 4. On page 8, there is a reference to "three core areas" but it then seems that four are listed. 5. Personally-speaking, I think it would be good to have a "Limitations" sub-title within the Discussion section. 6. Under 'Conclusions' in the Abstract, the authors state what will be done, in the future, under this heading. I think this section should present the conclusions Overall, though, a very good MS that should be published subject, I believe, to the above amendments.
--

VERSION 1 – AUTHOR RESPONSE

Reviewer 1 comments:

Abstract: The first line, 'Young people who have been removed from their family home and placed in out-of-home care have often experienced maltreatment', could have more impact. The first line of the article is much better: 'The most common reason a child would be removed from their home and placed in out-of-home care is due to the experience of abuse and/or neglect'

Thank you for your constructive feedback on this manuscript. We have now edited the first line of the abstract in line with your suggestions.

Key words: Suggest revision to include youth or adolescent, help-seeking.

Key words have been edited in line with suggestions. We have also now included the term 'help-seeking' more frequently throughout the main text.

Strengths and Limitations of the Study: Could be strengthened. What does this study add?

We have now added a final point on what this study adds, as well as edited the order of points.

Introduction

The final paragraph should be moved to methods. Also, can you please clarify if the participants were asked about their own experience or their views about young people generally.

Thank you for these suggestions. On page 7 (Methods) we have now clarified the types of questions we asked participants, which focused on their own experiences but also asked them to think about what advice they would give a friend. We have also now included the semi-structured interview schedule as supplementary material.

We have also revised the final paragraph, so it is more clearly about stating the aims of the study, rather than the methodology. The methodological information can then be found in the Methods section.

Methods: Can you be more specific about the decisions guiding sample size?

We have now added further information on the sample size (page 6). The main references we have used for justifications of sample size follow guidance from Vasileiou et al. (2018) and Marshall et al. (2015) – both of which are systematic reviews, specifically exploring the justification of sample sizes in qualitative research.

Page 6 line 29: 'urban' does not need to be capitalised.

Edited.

'Researcher characteristics and reflexivity' is listed as being included on page 7. This content could be much clearer.

Edited for clarity (page 8-9).

Results

The results seem to be very surface level (lacking depth) – relating to examples of positives and

negative.

We have edited the titles of some themes to improve clarity and have also edited the Results and Discussion for conciseness and clarity (main changes highlighted in red). We are hesitant to re-code the data, as the themes were generated via an extensive process that included two trained and experienced coders, the main one of whom was independent from the interviews (see page 8). The final codes were then also discussed at a consensus meeting. Thus, the process for generating the codes and themes was extensive, following Braun & Clarke guidance. The common split between positive and negative aspects of coping and support/help-seeking was also intentional. Rather than being surface level, this genuinely reflected the information provided by young people, who themselves tended to discuss things in terms of what worked and didn't work, as well as the common split that we saw between young people who were managing quite well in care versus those who were struggling with mental health difficulties (commented on throughout the Results; e.g., page 11; page 13; page 16).

We also believe that the inclusion of subthemes conveying positive information is important. Research with young people in care commonly focuses on negative aspects of experiences (understandably), but it was also important to convey that some participants reported feeling well supported and that being in care could be a positive experience (e.g., page 10). Crucially, our work with social-care and mental health services showed that this structure was also useful for them, in terms of their ability to easily interpret the findings in ways that are more useable for them to reflect on how they may need to change practice.

That said, we would be pleased to take further guidance from the Reviewer if they had suggestions of how they would prefer the Themes to be structured. We also hope our edited Discussion now better draws together our Results.

Discussion

It would be useful to consider strength-based and trauma-informed approaches to working with young people in care. The recommendations put forward by the authors seem at odds with what was recommended by participants.

We have now edited the Discussion to include a statement on strength-based and trauma-informed approaches, and also carefully edited further to ensure the Discussion is an appropriate reflection of the results (main changes highlighted in red). Reviewer 2 has also requested that we further embed our Discussion in the wider literature, so we have kept and expanded on some broader discussion related to the findings, particularly around the challenge of respecting the views of young people, but supporting them to engage in treatments that may require them to talk about their experiences (e.g., see page 19). This is an important consideration for trauma-informed approaches, although the answer is certainly not an easy one.

Page 20, line 20. 'Addressing these issues remains a significant challenge, but the current work found failing to provide adequate support, especially for those with the most significant mental health difficulties, had major consequences for the young person's wellbeing'. I'm not sure that the study supports this closing statement.

We have now edited this statement. This statement reflects our finding that young people with more significant mental health difficulties are often left feeling unsupported in placements and by services, creating a challenging cycle that can lead to worsening mental health.

Overall comments

While this group is under-researched, this study lacks depth in the results and entertains a particular lens of interpretation which was contrary to the views of participants.

We hope that we have addressed the reviewers concerns about ensuring the Discussion reflects the

views of the participants. As mentioned earlier, the coding was completed independently by two coders. The main coder had not been involved in the project or the interviews, to reduce potential bias interpretations (see page 8-9). We also used extensive reflexive practices wherever possible. Ensuring the voices of young people in care have been heard is central to the work conducted by our team, including via engaging with young people and services around study and interview design. We hope that our revisions accurately reflect this.

Reviewer: 2

The subject matter of this manuscript (MS) is important and interesting. Some of the findings are familiar, having been covered in previous literature, but they are no less important for this. This MS is focused upon a highly vulnerable group and it is vital to have current data on the welfare of these young people and the services they receive.

I think the MS is well written, being clear and concise, and it is well structured.

I had only one substantial criticism and that concerns the Discussion. I thought there was a bit too much of the findings in this section. There may be a bit of repetition from the Findings section or indeed new material that should have been in the Findings section. At the same time, I thought the Discussion section could have analysed more of the wider literature.

Thank you for your review of our paper. We have now edited the Discussion in line with your suggestions and the suggestions of Reviewer 1 (main changes highlighted in red text). This has included deleting approximately half a page of text that repeated the Results, and incorporating more discussion on the wider evidence base. We hope the edited Discussion now provides more of a balance between summarising our results and highlighting how they relate to the wider evidence base – both in terms of replicating findings from previous studies, and consideration of the broader practice and policy implications. We have also included more references and highlighted review papers, that readers can refer to for further information.

I had a number of smaller points:

1. 'Local authority/ies' and 'urban' are not proper names and should not be written with capitals.
2. The term "full care order" should be briefly explained for international readers.
3. I think "stake-holder" should not be hyphenated but should be a single word.
4. On page 8, there is a reference to "three core areas" but it then seems that four are listed.
5. Personally-speaking, I think it would be good to have a "Limitations" sub-title within the Discussion section.
6. Under 'Conclusions' in the Abstract, the authors state what will be done, in the future, under this heading. I think this section should present the conclusions

Overall, though, a very good MS that should be published subject, I believe, to the above amendments.

We have now edited all of the above points.

VERSION 2 – REVIEW

REVIEWER	Dr. Bernard Gallagher Independent writer and researcher UK
REVIEW RETURNED	03-Aug-2020

GENERAL COMMENTS	This is a generally well presented paper, being well-structured, clear and concise. This is an important topic and it is one that is under-researched. There are quite a few minor issues with this paper and I have highlighted these in the copy of the paper I am returning. There are larger issues I would highlight as follows:  1. Check whether it is appropriate to refer to self-harm as a coping strategy. 2. Check your assumptions and those of others regarding the benefits of young people talking about their abuse. 3. I believe there may be too much overlap between the Discussion and the Conclusions. Begin the Discussion with a short summary of your findings; structure your Discussion according to the main themes in your work and relate this to a possibly larger literature; and reserve any implications of/recommendations from your work (e.g. policy, practice and research) to the Conclusions sections. The Discussion section is not designed for recommendations but to help the reader to work out what to make of your research, as you compare it to the wider literature.  4. It would also be good to place your and related findings in a wider political context e.g. what is happening with CAMHS and austerity. I think this paper should be published. It is borderline minor-major amendments but i think the authors should be encouraged regarding what is essentially a very sound study and paper,
--

VERSION 2 – AUTHOR RESPONSE

Reviewer 2:

Edits provided in attached proof.

Thank you for highlighting that the paper is well presented, clear and concise, and that the topic is important.

We have endeavoured to address the suggested edits and comments that were on the Proof. We have edited words and accepted stylistic changes, as suggested. We have also given the manuscript another careful proof read for typographical errors.

To address specific comments:

Include further participant information against quotes:

As this is a relatively specific and small group of young people, recruited from only two English Local Authorities, we were uncomfortable having information that may make the young person possibly identifiable, even to their social worker or a current or past carer. To be extra cautious we have

provided the age of the young person, but no further details. The Editor has also advised that the journal prefers not to include the use of pseudonyms.

Consideration around shorter (6 minute) interviews:

We have now added the mean interview time. As is expected, young people varied in how verbose they were and what information they chose to provide. Some interviews were shorter because young people wanted to focus on one specific salient issue to them and some, of course, were just less verbose. As this is qualitative work, we have chosen to include all interviews so as not to make judgments about how important points were based on the length of the interview.

1. Check whether it is appropriate to refer to self-harm as a coping strategy.

We believe that it is appropriate to refer to self-harm as a maladaptive coping strategy. Coping strategies are ultimately not always helpful and can be key maintainers of poor mental health. Your comment that it may be a reaction to failing to receive support may very well be correct. In our Discussion, we have reflected on concerns that these young people often reported feeling as if they had no support, either from carers or professionals. Framing self-harm as a coping strategy is also in response to how young people discussed it. That is, in some cases (albeit the minority) young people reported that self-harm was the only way they knew how to cope with their thoughts and feelings. Some provided quite graphic depictions of how this “helped” them cope, but we have not included those quotes in text due to wider safeguarding considerations.

2. Check your assumptions and those of others regarding the benefits of young people talking about their abuse.

This is not an assumption, rather it is based on a large body of quantitative evidence from the trauma field that avoiding talking about trauma is associated with poorer long-term mental health outcomes (e.g., Trickey et al., 2012; Hiller et al., 2020; Mavranezouli et al., 2020). Of course, we are not implying that young people should be able to freely talk to everyone about what they have been through but rather that feeling in a safe space to talk about their experiences (e.g., to a caregiver or therapist) may be an important part of addressing poor mental health, particularly trauma-related mental health difficulties. On page 18 we have acknowledged the challenge here and been clearer that we are not implying young people should be forced to talk about their past. We have also clarified that some young people will not need to talk about their past. It is certainly a challenge for services that some young people emphatically do not wish to talk about their experiences, but we know that avoidance here is related to poorer outcomes. This is a challenge that we see in many groups of young people (and indeed adults) who have experienced trauma.

3. I believe there may be too much overlap between the Discussion and the Conclusions. Begin the Discussion with a short summary of your findings; structure your Discussion according to the main themes in your work and relate this to a possibly larger literature; and reserve any implications of/recommendations from your work (e.g. policy, practice and research) to the Conclusions sections.

The Discussion section is not designed for recommendations but to help the reader to work out what to make of your research, as you compare it to the wider literature.

4. It would also be good to place your and related findings in a wider political context e.g. what is happening with CAMHS and austerity.

Thank you for your suggestions on the Discussion. We appreciated that different academics and professions have different writing styles. We have edited the Discussion further and included a statement about underfunding of mental health services (page 20). We had also already referenced this issue in the Introduction. We believe the Discussion follows the guidelines of the journal and is

also consistent with how we have structured Discussions in previous publications, including a number of qualitative papers published in BMJ-Open. The opening paragraph provides a brief overview of findings, with following paragraphs embedding the findings in the wider literature, highlighting practice and policy implications (e.g., around access to mental health support, service policy around social worker changes), and making suggestions for future research. The conclusion then acts as a short finally summary of the paper. We had attempted to re-structure the Discussion to create a stand-alone policy and practice implications section, however this added significant length to the Discussion. Thus, we ultimately decided to leave the implications embedded within the Discussion. If however, the Editor feels the Discussion needs to be reworked and would be accepting of a longer version, we would be happy to go back and do so.

VERSION 3 – REVIEW

REVIEWER	Dr, Bernard Gallagher Independent researcher and writer Formerly, University of Huddersfield
REVIEW RETURNED	16-Nov-2020

GENERAL COMMENTS	This paper is based upon a modest, but still valuable, data set and the authors don't over claim in their paper. The authors have discussed a highly vulnerable group and have focused upon a key issue i.e. their mental health. Improvements in the conditions for, and lives of, children and young people in care are very slow in coming about and the pressure for this change must be kept up. Hence, this paper is very welcome. The paper is generally well written, being clear, concise and well structured. Overall, it is close to being ready for publication but it does need some minor amendments. The copy of the manuscript (MS) I am returning highlights the detailed amendments I would suggest. In addition, I would like to say:  1, I was surprised so little was said about these young people's peers i.e. as sources of support. A comment on this would be useful e.g. you didn't really ask about this or the yp did not have many friends or they appeared to seek out adults. 2. Another limitation might be that your self-selecting sample might have consisted of a disproportionate number of yp who were doing particularly well and/or those doing badly. 3. As you will see from the returned MS, I thought the organisation of some of the themes was a bit puzzling i.e. issues put together that were too distinct. 4. Some of the written English was awry in places i.e. grammar and punctuation. Apologies for the MS reformatting, only I had to open it Word to annotate it.
---

VERSION 3 – AUTHOR RESPONSE

Reviewer 2:

1. Comments on the manuscript:

Thank you for providing further comments within the manuscript. We have aimed to address these in our revised version. We have been able to address most comments including editing strengths and limitations and editing stylistic suggestions. While we have been able to address most of your comments, below you will find our rationale for some comments that we felt unable to address:
Request for rationale for exploring help seeking

We were unclear what information the reviewer was looking for here. As highlighted in the Introduction, the aim of the study was to explore young people's day to day coping with emotional difficulties. Naturally, this also includes exploring where young people might turn for help or support. We would be happy to add further information but further clarity would be useful around what further rationale the reviewer would like to see.

Add the approximate size of the local authority

We have chosen not to provide this. With information on the location of the authors and the size of the local authority, it would be very easy to extrapolate exactly what local authorities we are referring to, which risks participants being identified. Originally, as requested, we had added in age and sex of the participant against their quote. However, on further feedback from the journal Editor, we have reverted back to only having age against the quotes, to ensure participant identity is protected.

Information about the time between the interview and referrals to mental health services for support

Unfortunately, we do not have this information to report. While we supported social workers to refer young people for further support, via providing assessment summaries where appropriate, the research team were not involved in the formal referral and did not follow young people through this process, as it was not ethically appropriate.

Further dividing of sub-themes in Theme 1

Reflecting on the original reviews of the manuscript, including comments by the Editor and Reviewer 1, we have chosen not to further break-up the subthemes. The themes and subthemes were generated via a rigorous process that included two independent coders, including one who had not been involved in the study design or data collection. We are also conscious that the manuscript is already over the page limit of the journal, due to the qualitative quotes, and dividing subthemes would lengthen the manuscript further. That said, we have edited the titles of some subthemes so that they better reflect why the information has been grouped under a particular subtheme.

Of note, in thematic analysis sub-themes still involve the grouping of codes and information. Thus, it is acceptable that, while they are more specific than the overall theme, they contain different codes that are captured under one umbrella (in this case of Theme 1, positive aspects of being in care and negative aspects of being in care).

2. Comment on lack of focus on peer support

Information added to Discussion (page 18). We have also added an additional quote to the Results (page 12) to highlight that some older teens did indeed endorse friends as a primary source of support. We hope this is now clearer.

3. Comment on self selecting sample

This limitation is highlighted in the Discussion (page 20) and has been further edited in this version.

4. Theme organisation and grammatical issues

Please see above response for theme organisation. We have proof read the manuscript and aimed to correct grammatical issues. We expect the manuscript will also go through rigorous copy editing processes prior to publication.